Ketamine modulates subgenual cingulate connectivity with the memory-related neural circuit—a mechanism of relevance to resistant depression?

Wong Jing J.
O’Daly Owen
Mehta Mitul A.
Young Allan H.
Stone James M. james.m.stone@kcl.ac.uk
Institute of Psychiatry, Psychology and Neuroscience, King’s College London, University of London , London , United Kingdom
Patton Bob
Electronic publication date: 2016 Feb 16
Publication date: 2016
Volume: 4
Electronic Location ID: e1710
Received 2015 Oct 14; Accepted 2016 Jan 28
Copyright: ©2016 Wong et al.
Copyright year: 2016
Copyright holder: Wong et al.
License: This is an open access article distributed under the terms of the Creative Commons Attribution License, which permits unrestricted use, distribution, reproduction and adaptation in any medium and for any purpose provided that it is properly attributed. For attribution, the original author(s), title, publication source (PeerJ) and either DOI or URL of the article must be cited.
License URL: https://creativecommons.org/licenses/by/4.0/

Keywords: Ketamine, Depression, Anterior cingulate, Antidepressant, MRI

Funding: NIHR Biomedical Research Centre for Mental Health at the South London and Maudsley NHS Foundation Trust and Institute of Psychiatry, Psychology and Neuroscience, King’s College London This work was supported by the NIHR Biomedical Research Centre for Mental Health at the South London and Maudsley NHS Foundation Trust and Institute of Psychiatry, Psychology and Neuroscience, King’s College London. The funders had no role in study design, data collection and analysis, decision to publish, or preparation of the manuscript.

==============================
Background. Ketamine has been reported to have efficacy as an antidepressant in several studies of treatment-resistant depression. In this study, we investigate whether an acute administration of ketamine leads to reductions in the functional connectivity of subgenual anterior cingulate cortex (sgACC) with other brain regions.

Methods. Thirteen right-handed healthy male subjects underwent a 15 min resting state fMRI with an infusion of intravenous ketamine (target blood level = 150 ng/ml) starting at 5 min. We used a seed region centred on the sgACC and assessed functional connectivity before and during ketamine administration.

Results. Before ketamine administration, positive coupling with the sgACC seed region was observed in a large cluster encompassing the anterior cingulate and negative coupling was observed with the anterior cerebellum. Following ketamine administration, sgACC activity became negatively correlated with the brainstem, hippocampus, parahippocampal gyrus, retrosplenial cortex, and thalamus.

Discussion. Ketamine reduced functional connectivity of the sgACC with brain regions implicated in emotion, memory and mind wandering. It is possible the therapeutic effects of ketamine may be mediated via this mechanism, although further work is required to test this hypothesis.

Introduction

The subgenual anterior cingulate (sgACC), or Brodmann area 25, is a brain region that has been implicated in the control and modulation of mood. Decreased mean sgACC gray matter volume combined with metabolic hyperactivity and hyperconnectivity has been observed in patients with major depressive disorder (MDD) (Davey et al., 2012; Drevets, Savitz & Trimble, 2008; Greicius et al., 2007; Sundermann, Olde Lutke Beverborg & Pfleiderer, 2014). With a growing number of studies demonstrating the association between metabolic hyperactivity in this region and poor therapeutic response (Baeken et al., 2014; Konarski et al., 2009; Maletic & Raison, 2009; Sheline et al., 2010; Taylor & Liberzon, 2007), the activity of this region may prove important for attempts to predict treatment response in patients (Siegle et al., 2012). Furthermore, given that connectivity between the sgACC and the default mode network (particularly the ventromedial prefrontal cortex, and posterior cingulate cortex) has been hypothesised to underlie depressive rumination (Hamilton et al., 2015), therapeutic disruption of this connectivity might also be a potential target for novel antidepressant action.

Treatment resistant depression (TRD) is common, with approximately 45% of patients failing to respond to pharmacological treatment (Papakostas & Fava, 2010). Currently available antidepressants are characterised by a relatively slow onset of effect ranging from weeks to months. There has been great interest in the potential of ketamine as a treatment for TRD patients, given early reports of strong and rapid (within hours) antidepressant properties in patients otherwise resistant to antidepressant treatment (Fond et al., 2014; Salvadore et al., 2009; Zarate et al., 2012; Zarate et al., 2006).

Ketamine has been reported to reduce sgACC activity in healthy volunteers (De Simoni et al., 2013; Deakin et al., 2008; Doyle et al., 2013; Stone et al., 2015), and responders to ketamine with treatment resistant bipolar depression have been reported to have increased glucose metabolism in this region (Nugent et al., 2014). However, the effect of ketamine on subgenual connectivity, which might be hypothesised to underlie its antidepressant effects, has not been fully investigated. To date there has only been one study on the effect of ketamine on brain connectivity in relation to potential antidepressant mechanisms. Ketamine was shown to disrupt connectivity between the sgACC and the dorsal nexus 24 h following administration in healthy volunteers, an effect that was hypothesised to be related to its antidepressant properties (Scheidegger et al., 2012). We hypothesise that immediate effects of ketamine might also play an important part in the therapeutic effects of ketamine in TRD, and that the early modulation of circuits involved in maintenance of depressive cognitions may be necessary for the emergence of objectively measureable clinical improvement.

In this study, which is an analysis of existing resting state data (Stone et al., 2015), we investigated the effect of acute intravenous ketamine administration on functional connectivity of the sgACC in healthy volunteers.

Materials and Methods

The study was approved by the East London Research Ethics Committee. Prior to screening for the study, all participants gave written informed consent for inclusion. Inclusion criteria: healthy right-handed male subjects (18–50 years old) with a total body weight of 50–100 kg. Exclusion criteria: positive urine drug screen for drugs of abuse, the consumption of more than 5 cups of coffee per day, smoking more than 5 cigarettes per day, taking prescription drugs, and any history of mental illness, or serious medical condition that, in the opinion of the study doctors, prevented their participation in the study.

Thirteen healthy, right handed, male volunteers (age 21–39, mean 27, standard deviation 6.90) were selected for the study after screening. Each subject underwent medical, mental state, physical, and psychiatric examination including electrocardiogram, urine drug screen, and taking measurements of blood pressure, pulse rate, temperature, and weight. Each volunteer underwent venous cannulation in the left antecubital fossa. We attached a 50 ml syringe pump containing 4 mg/ml racemic ketamine via an infusion line.

Image acquisition was conducted, as previously reported (Stone et al., 2015), at the Centre for Neuroimaging Sciences on a General Electric (Milwaukee, Wisconsin) 3-Tesla HDx MRI scanner. Pharmacological MRI (phMRI) blood-oxygen level dependent (BOLD) data were acquired using gradient echo EPI (Echo-Planar Imaging) with parallel imaging accelerated by a factor of 2. Each participant was scanned continuously for 15 min to yield a total of 450 functional image volumes of 37, continuous top down, 3 mm thick slices with a slice gap of 0.3 mm, repeat time (TR) of 2000 ms, echo time (TE) of 30 ms, flip angle of 75°, in-plane resolution of 3.3 mm, 64 × 64 matrix, and 21.1 × 21.1 cm field of view. The ketamine infusion commenced after 5 min of resting state acquisition and followed a dynamically modelled intravenous infusion with a target plasma level of 150 ng/mL determined according to pharmacodynamic properties of ketamine from the “Clements 250 model”, with a rapid bolus over 20 s of approximately 0.26 mg/Kg, on average, followed by a slow infusion of approximately 0.42 mg/Kg/Hr. Participants’ peak ketamine-induced experience was rated by a trained psychiatrist using the positive and negative syndrome scale (PANSS) immediately following their exit from the scanning room.

Preprocessing and statistical analyses were performed using Statistical Parametric Mapping software version 8 (SPM8; Wellcome Trust Centre for Neuroimaging, London, England). Functional images were corrected for slice timing effects and subsequently realigned to correct for the effects of volume-to-volume head motion. Data were examined to confirm that framewise translational and rotational head movement did not exceed 2 mm or 1°. Images were co-registered to a high-resolution T1-weighted structural image, and normalised to Montreal Neurological Institute (MNI) space via unified segmentation. The normalised images were smoothed using an 8 mm full width at half maximum (FWHM) Gaussian kernel. Additional preprocessing was carried out using the REST toolbox for resting state fMRI analysis (Song et al., 2011). Nuisance variables such as motion parameters (volume to volume translations in 3 axes and rotation around these axes), white matter and CSF signal were regressed from the data. The residual time-series was then de-trended and band-pass filtered (frequency range 0.01–0.08 Hz) and a signal time-series was extracted from the sgACC seed (10 mm sphere at [2, 28, −5] based upon a previous publication (Scheidegger et al., 2012)). The 15-minute time series was separated into three 5-minute time-series segments of pre-infusion, early-infusion and late-infusion. The early-infusion portion of the time series was disregarded as any observed connectivity would have been dominated by the phMRI response to the bolus. Finally, connectivity maps between the sgACC seed and the whole brain were created using regression within the REST toolbox, and the resultant r-maps underwent r-to-Z conversion again within REST.

These Z-transformed maps were taken forward into a second level random effect analysis within SPM8. A one sample t-test was used to characterise sgACC connectivity prior to ketamine administration, and a paired t-test was employed (comparing pre- and post-ketamine connectivity) to identify regional changes in sgACC coupling following ketamine administration. We also directly investigated connectivity between the sgACC seed and the dorsal nexus region, by use of two spherical masks of 10 mm radius at x = ± 36, y = 27, z = 29 (Sheline et al., 2010). Results were considered significant if they survived family wise error (FWE) correction on the basis of cluster-extent (pFWE < 0.05). The PANSS general score was tested for normality using the Shapiro–Wilk test prior to regression analysis against ketamine-induced change in sgACC connectivity.

Results

Prior to ketamine administration, there was positive coupling (pFWE < 0.05) between sgACC and multiple brain regions including anterior cingulate, ventral striatum, and thalamus. There was negative coupling (pFWE < 0.05) between sgACC and regions including cerebellum, pons, precentral gyrus, superior frontal gyrus, and parahippocampus (Table 1).

Following ketamine administration, there was significant reduction in sgACC coupling with a large cluster including the hippocampus, retrosplenial cortex (RSC), and thalamus centred at [−2 −3 6] (pFWE = 0.002; kE = 2,885; Zo = 3.69) (Fig. 1). Plotting the contrast estimate from the cluster peak, located in the retrosplenial cortex [−6 −55 3], revealed that the RSC was uncorrelated with the sgACC before ketamine administration, but that it was negatively correlated with the sgACC following ketamine administration (Fig. 2). There was no significant change in connectivity between sgACC and the dorsal nexus region following ketamine administration.

Table 1 Resting state sgACC coupling prior to ketamine administration.

Regions showing significant coupling with sgACC prior to ketamine infusion (pFWE < 0.05 corrected for multiple comparisons on the basis of cluster extent, using a cluster-forming threshold of z = 3.1).

sgACC coupling	Brain region	Brodmann area	Cluster-level	Peak-level	Coordinates	
			pFWE	kE	(Zo)	x	y	z	
Positive	Ventral Anterior Cingulate	24	<0.001	12,610	6.69	4	27	−6	
Positive	Dorsal Anterior Cingulate	32			6.23	−6	35	−3	
Positive	Thalamus				5.79	−2	−3	−3	
Negative	Anterior Cerebellum		<0.001	3,705	4.69	−12	−37	−27	
Negative	Pons				4.16	9	−22	−24	
Negative	Anterior Cerebellum				3.96	10	−37	−26	
Negative	Middle Frontal Gyrus	9	<0.001	1,089	4.68	43	12	33	
Negative	Middle Frontal Gyrus	9			4.50	42	3	39	
Negative	Precentral Gyrus	6			4.03	51	0	40	
Negative	Superior Frontal Gyrus	10	0.001	971	4.45	22	51	24	
Negative	Middle Frontal Gyrus	9			4.12	34	29	28	
Negative	Middle Frontal Gyrus	9			3.77	30	36	21	
Negative	Inferior Parietal Lobule	40	0.007	649	4.17	66	−46	31	
Negative	Inferior Parietal Lobule	40			3.46	70	−46	22	
Negative	Postcentral Gyrus	2			3.44	61	−30	43	

Figure 1 sgACC connectivity following ketamine administration.

Regions showing significant (pFWE < 0.05) reduction in sgACC coupling following ketamine administration (red/yellow).

Figure 2 Effect of ketamine on connectivity between sgACC and retrosplenial cortex.

Correlation between sgACC and retrosplenial cortex [−6 −55 3] before and after the start of ketamine administration.

Ketamine administration was associated with a mean (SD) increase in PANSS positive, negative, and general subscales to 10.7(2.89), 10.07(3.43), and 20.15(3.53) respectively (p < 0.05). Multiple regression analysis using cluster forming threshold of p < 0.01 was performed to test for correlations between changes in whole brain functional connectivity with sgACC following ketamine administration and PANSS scores. No correlations with PANSS positive or negative scores were found, but a negative correlation between the PANSS general score and sgACC coupling was observed in the medial prefrontal cortex (mPFC) and subcallosal gyrus (SCG) (pFWE < 0.05). In order to further investigate this correlation, we performed a post-hoc analysis of the correlation between sgACC coupling and depression-related items using the 5 factor PANSS (Lindenmayer, Bernstein-Hyman & Grochowski, 1994). The level of the depression subscale (primarily anxiety and preoccupation rather than the depression item) was significantly increased following ketamine administration (p < 0.05), and was found to be negatively associated with coupling between sgACC and subcallosal gyrus and with right ventrolateral prefrontal cortex (pFWE < 0.05). The depression subscale was positively associated with coupling between sgACC and right ventromedial prefrontal cortex (pFWE < 0.05; Fig. 3).

Figure 3 Correlation between PANSS depression and sgACC coupling.

Regions showing significant (pFWE < 0.05) correlations between PANSS depression score and change in sgACC coupling following ketamine administration (blue—negative correlation, yellow—positive correlation).

Discussion

In this study, we examined the acute effect of ketamine on functional connectivity between the sgACC and other brain regions. Our primary aim was to investigate the effect of acute ketamine administration in healthy volunteers on brain networks hypothesised to be involved in the aetiopathology of depression. The most striking effect of acute ketamine administration in this study is the disruption of connectivity between sgACC and a large cluster encompassing midline thalamus, hippocampus, RSC. This may be of relevance to the antidepressant effect. Both thalamus and hippocampus have been implicated in the pathology of MDD (Malykhin & Coupland, 2015; Yakovlev et al., 1960; Young et al., 2004). Furthermore, the network connectivity between sgACC and the default mode network, including RSC is increased in patients with MDD, and has been suggested to underlie depressive ruminations (Hamilton et al., 2015), a process hypothesized to be of great significance in the maintenance of depressed mood (Disner et al., 2011). If ketamine is able to disrupt the tendency of the mind to return to depressive ruminations through changing the functional connectivity within this network, this may be of particular importance in its antidepressant action. Interestingly, two studies of ketamine effects on functional connectivity revealed an increase rather than a decrease in connectivity between hippocampus and prefrontal cortex following ketamine administration (Gass et al., 2014; Grimm et al., 2015). In addition a graph theory analysis of whole brain connectivity showed a pattern of increased subcortical/cerebellar connectivity after ketamine in healthy volunteers (Joules et al., 2015). It is worth noting, however, that these studies did not specifically examine connectivity with sgACC, and it is possible that a decrease in connectivity may have been seen between this region and the hippocampus.

Although ketamine acutely increased rather than decreased scores on the 5 factor PANSS depression scale in this healthy volunteer sample (it would not have been possible to for depressive scores to reduce as volunteers did not have any symptoms before ketamine administration) it is notable that changes in functional connectivity between subgenual anterior cingulate and other brain regions correlated only with the general and depressive scores on the PANSS, and not with positive or negative subscales. Our finding of a correlation between ketamine-induced increases on the depression subscale and reduced sgACC coupling with surrounding regions of the SCG is of interest as the SCG is an important node in a neural network comprising of cortical structures, the limbic system, thalamus, hypothalamus, and brainstem nuclei. MDD patients generally show increased activity in SCG, which is normalised by antidepressant treatment (Hamani et al., 2011). It is possible that this local reduction in connectivity reflects a direct consequence of the reduced BOLD activity in this brain region seen following ketamine administration (Stone et al., 2015).

It is interesting that we found a reduction in connectivity between sgACC and ventrolateral prefrontal cortex associated with increases in ketamine-induced scores on the depression subscale, whereas there was an associated increase in connectivity between sgACC and ventromedial prefrontal cortex connectivity with the same scale. The reason for this is not entirely clear, but both ventrolateral and ventromedial prefrontal cortices have been implicated in depression (Ma et al., 2013; Miller et al., 2015). Ventromedial prefrontal cortex connectivity with sgACC has been suggested to underlie depressive ruminations (Hamilton et al., 2015), so the fact that ketamine acutely increased depression scores and increased this connectivity is in keeping with this finding. In contrast, one study in patients with treatment resistant unipolar depression found increases in connectivity between anterior cingulate cortex and a cluster including ventrolateral prefrontal cortex associated with clinical response to transcranial magnetic stimulation (Baeken et al., 2014), and thus decreases in connectivity between these two regions might be predicted to increase depressive symptoms. However, it is not at all clear how ketamine-induced increases in depression scores in healthy volunteers may be related to its effect as an antidepressant in patients with existing clinical depression.

There are a number of limitations regarding the study design. The fact we were studying healthy volunteers means that any effects we see in functional connectivity may not map directly onto those that occur in patients with MDD. Ketamine modulation of brain circuits may vary according to severity of depression and thus networks affected in healthy controls may be different to those suffering from MDD. Related to this, we were not able to measure the antidepressant effect of ketamine as the volunteers had no measureable depressive symptoms at presentation, and so the current data cannot provide evidence that ketamine-induced changes in connectivity are related to ketamine’s antidepressant effect. Secondly, the effects investigated in the present study occurred 5 min following ketamine administration, during the steady-state infusion, whereas antidepressant effect in patients do not normally arise until 40 min to 2 h after ketamine administration. We hypothesise that the changes reported in the current study may be precursors to an antidepressant effect, but it is possible that changes in functional connectivity relevant to antidepressant mechanism between other brain regions might arise at later time points. Lastly, we did not have a placebo arm. However, participants were not aware when the ketamine was started, and our earlier placebo-controlled study using an identical method of ketamine infusion did not show any effect of placebo on BOLD signal in any brain region (Doyle et al., 2013).

Conclusion

We found that ketamine alters the functional connectivity of the sgACC in healthy controls. The networks affected suggest that these changes may be of importance in the therapeutic effects of ketamine in patients with MDD. Further studies in patients are required to test the hypothesis that ketamine may reduce depressive rumination via acute effects on sgACC- RSC connectivity.

Supplemental Information

Supplemental Information 1 3D nifti file format output data

Click here for additional data file.

Additional Information and Declarations

Competing Interests

Author Contributions

Human Ethics

Data Availability

The authors declare there are no competing interests.

Jing J. Wong analyzed the data, wrote the paper, prepared figures and/or tables, reviewed drafts of the paper.

Owen O’Daly conceived and designed the experiments, analyzed the data, contributed reagents/materials/analysis tools, wrote the paper, reviewed drafts of the paper.

Mitul A. Mehta conceived and designed the experiments, analyzed the data, contributed reagents/materials/analysis tools, wrote the paper, prepared figures and/or tables, reviewed drafts of the paper.

Allan H. Young wrote the paper, reviewed drafts of the paper.

James M. Stone conceived and designed the experiments, performed the experiments, analyzed the data, wrote the paper, prepared figures and/or tables, reviewed drafts of the paper.

The following information was supplied relating to ethical approvals (i.e., approving body and any reference numbers):

The study was approved by the East London Research Ethics Committee.

The following information was supplied regarding data availability:

Associated data are uploaded as Supplemental Information.

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
