# Peer review of "Ketamine modulates subgenual cingulate connectivity with the memory-related neural circuit—a mechanism of relevance to resistant depression?"

_PeerJ, doi:10.7717/peerj.1710_

## Round 0.1 · original submission · Major Revisions

Please address the concerns of both reviewers, in particular those relating to the methodology. Consider your inclusion of Serotonin in the introduction, or make appropriate modification to justify its inclusion. Your discussion should also include comments around the limitations of a lack of placebo control.

Reviewer 1 ·

Basic reporting

The inclusion and exclusion criteria used to arrive at this the sample should be reported.

The abbreviation "RSC" is not defined.

Experimental design

The approach used to address motion in the scanner should be reported. Did the authors examine the data to characterize the motion, eliminate or de-weight volumes that exceeded a threshold, and exclude subject data that had too much motion?

Validity of the findings

In the introduction, authors state: "In this study, which is an analysis of existing resting state data (Stone et al. 2015), we investigated the effect of acute intravenous ketamine administration on functional connectivity of the sgACC in healthy volunteers."

I think this statement is an appropriate goal that fits with their study design.

In contrast, in the discussion they state: "Our primary aim was to investigate
potential mechanisms underlying the antidepressant effect of ketamine."

I disagree with this. I do not think the study design was suited to carry out the aim of examining antidepressant mechanisms of ketamine, since the sample did not have depression.

A limitation to the study design that was not discussed is the lack of placebo control. Resting state functional connectivity is dynamic, and there may be changes that occur over time in the absence of ketamine. A drug effect can only be proved either with a placebo control, or perhaps using a dose-response study design.

Additional comments

Overall the paper is well written. I was surprised that the authors began the introduction with discussing serotonin, which is not relevant to their study and not mentioned anywhere else in the paper.

Reviewer 2 ·

Basic reporting

Generally well written paper, however there are areas which lack sufficient detail and clarity. Please see individual sections for specific issues.

Minor comment:
• Abbreviations need to be defined first before used in the text.

Experimental design

• There is insufficient detail of the connectivity analyses carried out using REST. For those not familiar with the software, it is difficult to understand exactly what was carried out. You state that signal time-series were extracted from the sgACC seed but do not state how these were entered into the FC analysis (i.e. was this a whole brain search, what masks if any, were used?)
• The statistical analysis is also not clear enough – what test was specifically used for the random effects analysis?
• It is also not clear whether the correlations are looking at “change” in connectivity, or the connectivity after ketamine administration (and this is not detailed in the methods). It appears in the results as though correlations were run just for the post ketamine connectivity maps. Perhaps it would be of more interest to see if difference maps correlate with change in mood?
• I am also puzzled as to why “symptoms” were measured at all since a) they are healthy subjects, it doesn’t seem appropriate to try to correlate these with connectivity (the effects will probably be very small in any case and likely run into a floor effect), and b) in patients, mood effects aren’t seen until around 1 hour so why would you expect any change in healthy controls?
• Was the increase in symptom scores significant?
• On line 132 – what is meant by “An examination of the model coefficients”. Were means extracted from the cluster? How are you defining “strong correlation”.
• On line 133 – “at rest” should be before ketamine?
• In general, the analysis could be greatly improved by focusing on specific networks that are implicated in depression (i.e. those that are showing up in the pre-ketamine connectivity analysis), then seeing if the connectivity in these networks change following ketamine. As it stands, I am not sure that it really adds much to the literature in terms of ketamine treatment for depression. However, the changes in the hippocampal regions are interesting, and noteworthy, which could be explored much further, especially given that hippocampus connectivity has shown effects following ketamine administration in rats for example.

Validity of the findings

• In relation to collecting PANSS data, care needs to be taken in the interpretation as “depressive symptoms” that is referred to in the text would be very different to depressive symptoms of a patient suffering from TRD, and therefore would be represent very different neural substrates. There is no evidence with the current data that the connectivity change due to ketamine is related to an antidepressant effect.
• The correlations seen with the PANSS scores are interesting (although I do believe this needs to be done more appropriately). The DLPFC and DMPFC connectivity with sgACC show opposing patterns of relationship with symptom scores – this is intriguing. I would expect some discussion on this especially given these areas are heavily implicated in depression literature (e.g. Sheline’s dorsal nexus).
• On line 159, what is the relevance of the “retrieval of fear memory”?
• On line 161, what is increased? Connectivity needs to be included here.
• On line 169, floor effect is perhaps not an appropriate reason here since scores increased (rather than not changed).
• On line 176 – I wonder if this notion could be tested in some way with the current data. Perhaps using DCM or mediation?

Additional comments

Summary: The study investigated the acute effects of ketamine in healthy controls on the functional connectivity of the sgACC based on reports of the effectiveness of ketamine for treating treatment-resistant depression. The main finding was a reduction in connectivity of the sgACC with memory-related regions such as the hippocampus and surrounding regions.

The paper is generally written well. The novelty of this study is in the findings of ketamine affecting the connectivity of the sgACC with the memory-related network, which is interesting and of value for the ongoing investigations into the use of ketamine for depression treatment. However, there are some weaknesses in the manuscript which relate mainly to the methods which are highlighted below. On improving the analysis, I believe this manuscript would be interesting to the readers, however as it stands, it needs substantial revisions before being accepted for publication.

---

## Round 0.2 · accepted · Accept

Thank you for your re-submission. Following a second review, we are please to accept your paper for publication.

Reviewer 2 ·

Basic reporting

No comments

Experimental design

No comments

Validity of the findings

No comments

Additional comments

The manuscript is now much improved following the revisions. I recommend that this paper be accepted for publication.